

# Prebiotic supplementation effect on *Escherichia coli* and *Salmonella* species associated with experimentally induced intestinal coccidiosis in rabbits

Shawky M. Aboelhadid[1,*], Shaymaa Hashem[2], El-Sayed Abdel-Kafy[2], Lilian N. Mahrous[1], Eman M. Farghly[2], Abdel-Azeem S. Abdel-Baki[3], Saleh Al-Quraishy[4] and Asmaa A. Kamel[1,*]

[1] Parasitology Department, Faculty of Veterinary Medicine, Beni-Suef University, Beni-Suef, Egypt
[2] Animal Production Research Institute, Agricultural Research Center, Dokki, Giza, Egypt
[3] Zoology Department, Faculty of Science, Beni-Suef University, Beni-Suef, Egypt
[4] Zoology Department, College of Science, King Saud University, Riadh, Saudi Arabia
* These authors contributed equally to this work.

Corresponding author
Shawky M. Aboelhadid,
shawky.abohadid@vet.bsu.edu.eg

## ABSTRACT

**Background**. Coccidian infection may enhance the proliferation of gut Enterobacteriaceae. Bacterial infections in rabbits can negatively affect the body condition and cause high mortality, especially at young ages. Therefore, the effect of prebiotic supplementation on the presence of *Escherichia coli* and *Salmonella* species in rabbits experimentally infected with intestinal coccidiosis was investigated.

**Methods**. Thirty male rabbits aged 35–40 days were divided into three equal groups. These groups were; prebiotic supplemented (PS), positive control (PC), and negative control (NC) groups. The prebiotic group was supplemented with 2 g/L of Bio-Mos® until the end of the experiment. At day ten post prebiotic supplementation; the PS and PC groups were inoculated orally with $5.0 \times 10^4$ sporulated oocysts of mixed species of rabbit *Eimeria*. The daily fecal examination was carried out from the day 4 post-infection (PI) until the day 8 PI. At day 5 and day 8 PI, 5 rabbits from each group (PS, PC, and NC) were humanely slaughtered and parts of intestinal tissue were collected for microbiological analysis.

**Results**. There was a significant decrease ($P \leq 0.05$) in the oocyst count in the PS group ($25.12 \times 10^4 \pm 10.36$) when compared with the PC group ($43.43 \times 10^4 \pm 11.52$) and this decrease was continued till the end of the experiment. Eleven *E. coli* isolates were detected in the collected samples with an overall prevalence of 24.4%. The highest prevalence of *E. coli* was in the PC group (13.33%) while the lowest one was in the PS group (4.44%). Meanwhile, four *Salmonella* serovars were isolated with an overall prevalence of 8.89%. The NC group showed one serovar (2.22%) and PC revealed three serovars (6.67%) while the prebiotic supplemented group didn't show any *salmonella* isolate. Of *E. coli* isolates, five isolates (O78, O125, O152, O115 and O168) showed high resistance to florfenicol and neomycin (100%). Also, of *salmonella* serovars, thee serovars (*Salmonella entrica* subsp. *enterica* serovar Macclesfield, *Salmonella entrica* Subsp. *enterica* serovar Canada and *Salmonella entrica* Subsp. enterica serovar Kisangani) showed high resistance to sulphamazole, amoxicillin

and flumequin (75%) while it was sensitive to levofloxacin and ciprofloxacine (75%). The bacterial colony in this study was the same results at days 5 and 8 PI.

**Conclusion**. The use of prebiotic as prophylaxis in this experiment significantly reduced the prevalence of *E. coli* and *salmonella* associated with the intestinal coccidiosis in rabbits.

# INTRODUCTION

Nowadays, the antibiotic resistance has emerged as the greatest challenge in the animal production and human health. The extensive use of antibiotics as growth promoters for livestock is the major cause of antibiotic resistance (*World Health Organization & Communicable Diseases Cluster, 2000*; *Millman et al., 2013*; *Mitchell et al., 2013*). Rabbit is a good source of animal protein to fill the gap of the red meat shortage in some parts of the world, and sometimes reared for fur production as well as for medical and biological purposes (*Dalle Zotte & Szendro, 2011*; *Aboelhadid et al., 2019*). Rabbit coccidiosis is caused by apicomplexan parasites of the genus *Eimeria* (*Pakandl, 2009*). Coccidiosis is mainly occurring in young rabbits of one to three months' age especially after weaning. The clinical signs of coccidiosis are: reduced appetite, enteritis, diarrhea, and in severe cases infection may result in death (*El-Shahawi, El-Fayomi, Abdel-Haleem, 2011*)). There are two types of rabbit coccidiosis; intestinal coccidiosis which caused by several species including *E. intestinalis*, *E. perforans*, *E. magna*, *E. media*, and *E. irresidua* and, hepatic coccidiosis which caused by only *E. steidae* (*Pakandl, 2009*). *Rashwan & Marai (2000)* postulated that the coccidian infection may enhance the proliferation of the gut Enterobacteriaceae. Bacterial infections in rabbits can negatively affect the body condition and cause high mortality, especially in young ages (*Zahraei, Mahzouniehand & Khaksar, 2010*). Infections with Enterobacteriaceae are more challenging to treat, because few, and in some cases no, antimicrobials remain effective against them, because of their extensive resistance patterns and in addition the antibiotic chemical residue in animal products may create problems for human wellbeing (*Smith et al., 2002*).

*Salmonella* species were reported to infect rabbits in several rabbitries and the infection can lead to severely diseased condition (*Agnoletti et al., 1999*; *Zahraei, Mahzouniehand & Khaksar, 2010*; *Borrelli et al., 2011*).

*Escherichia coli* is an important cause of diarrhea in both animals and humans (*Nguyen et al., 2006*; *García, Fox & Besser, 2010*). Also, it was reported to cause morbidity and mortality in large laboratory rabbits (*Swennes et al., 2012*; *Swennes et al., 2014*). *Escherichia coli* was known as a reason for diarrhea in new born of New Zealand rabbits (*Camarda et al., 2012*; *Hamed, Eid & El-Bakrey, 2013*). *Prescott (1978)* found that the outbreak of severe diarrhea and death in young rabbits was associated with non-enterotoxigenic *Escherichia coli* (O153).

Recently, the prebiotics were defined as "a substrate that is selectively utilized by host microorganisms conferring a health benefit" (*Gibson et al., 2017*). The prebiotics administration could regulate specific gastrointestinal tract microorganisms to modify the microbiome (*Gibson & Roberfroid, 1995*). *Abdelhady & El-Abasy (2015)* found that the prebiotic and probiotic as dietary supplementation reduced the mortality rate and improved the adverse clinical signs of *Pasterella multocida* in experimentally infected rabbits. *Tzortzis et al. (2005)* realized that the oligosaccharides greatly inhibited the adhesion of *E. coli* and *Salmonella* to HT29 cells. Also, *Yusrizal & Chen (2003)* revealed that fructans supplementation induced an increase in *Lactobacillus* bacteria and a reduction in *Salmonella* in the broiler chickens. Moreover, it was noticed that prebiotics intake reduced the establishment of *Salmonella* in the course of hen molting (*Donalson et al., 2008*). *Spring et al. (2015)* demonstrated that Bio-Mos®, which has been used in the animal husbandry industry, plays a crucial role in animal nutrition and production. It was extracted from a selected strain of *Saccharomyces cerevisiae* yeast. Bio-Mos is inserted into animal diets to support overall animal performance and rapid growth. It is supported by over 734 trials and 114 peer-reviewed publications. There is considerable evidence now that Bio-Mos is among the best alternatives to antibiotic and growth promotants (*Ferket, Parks & Grimes, 2002*).

The present study was therefore conducted to explore the effect of a prebiotic supplement as a prophylaxis on the presence of *E. coli* and *Salmonella* species in rabbits with experimentally induced intestinal coccidiosis.

## MATERIALS AND METHODS

This experiment was conducted under the roles of the ethical standards approved by Faculty of Veterinary Medicine, Beni-Suef University, Egypt and its specific approval number was (BSUV-39/2019).

### Rabbits

A total of thirty male rabbits recently weaned (V-Line breed) aged 35–40 days with an average weighed of one kg, were used in the current work. The experiment was carried out in a rabbit farm in a station of animal production in Sedes station for agriculture research, Beni-Suef, Egypt. The rabbits were housed in metal cages where a single rabbit was located in a separate cage. Rabbits fed on anticoccidial drugs free commercial pelleted diet (18% crude protein, 14% crude fiber, 2,500 k calories digestible energy /kg, 1% calcium and 0.5% phosphorus). The water and feed were *ad libitum*. The fecal examination was done daily before induction of the infection to be sure that the rabbits were free from any other parasites.

### Experimental design of prebiotic prophylaxis efficacy

The thirty rabbits were randomly divided into 3 groups with10 rabbits in each group. These groups were as following: Prebiotic supplemented group (PS), positive control group (PC), and negative control group (NC). The prebiotic group was supplemented with 2 g/L of prebiotic (Bio-Mos®, ALLTECH, INC. CO. USA) derived from a selected strain

of *Saccharomyces cerevisiae* yeast in the drinking water for ten days while the other groups remained as they were. At day ten post prebiotic supplementation; the PS and PC groups were orally inoculated with $5.0 \times 10^4$ sporulated oocysts of mixed *Eimeria* species including *E. media, E. flvescens, E. intestinalis* and *E. magna* for each rabbit. The supplementation of prebiotic in PS group continued till the day 8 post infection. Daily fecal examination and oocysts count were carried out from day 3 until day 8 post infection (PI). The oocyst count was done using McMaster chamber (*Lillehoj & Ruff, 1987*). At the day 5 and the day 8 PI, 5 rabbits from each group at each period were humanely slaughtered. These days was selected based on a preliminary work in which the oocyst shedding started at day 5 PI and reached to its peak at day 8 PI as shown in the supplemented figure. Parts of the intestinal tissue were excised for microbiological analysis. These parts were labeled and kept in ice tank then rapidly transported to the laboratory for examination. The rabbits were handled and euthanized with least distressful to them. Cervical dislocation was done because they were not of heavy weight (*Walsh, Percival & Turner, 2017*). Death was verified by lack of breathing, stop palpable heartbeat and fixed dilated pupil.

## Bacteriological examination

The intestinal tissue samples (jejunum, ileum, and cecum) were collected separately in sterile manner for each point of microbial investigation at 5 and 8 days PI. Consequently, a total 45 samples representing 15 animals at the selected days, 5 animals from each group and three organs (jejunum, ileum, and cecum) for each animal. These samples were subjected to microbiological examination for the presence of *Salmonella and E. coli*. Two intestinal swabs were taken from each part. The first swab was seeded onto MacConkey Agar (Difco) to isolate *E. coli*, and the subsequent colonies were recognized using Enterokit B and identified according to *Lee et al. (2009)*. To isolate *Salmonella*, the second swab was processed according to *Michael et al. (2003)*. The produced colonies were confirmed according to the standard procedures suggested by *Holt et al. (1994)*, *ISO 6579 (2002)* and *Lee & Arp (1998)*. *Salmonella* isolates were serologically identified referring to somatic (O) and flagellar (H) antigens by slide agglutination using commercial antisera (*Popoff & Le Minor, 2001*).

## Antimicrobial susceptibility test

The disk-diffusion method was applied to assess the antibiogram of the isolated microorganisms (*Cruickshank et al., 1975*, *CLSI/NCCLS, 2009*) against a series of 12 antibiotic discs (Tetracycline, Sulphamazole, Naldixic acid, Trimethoprim, Gentamycin, Levofloxacin, Florfenicol, Amoxicillin, Flumequin, Ciprofloxacine, Amikacin, Neomycin) (Oxoid, Basingstoke, UK) (Table S1).

## Statistics

ANOVA tests and subsequent Duncan's multiple range tests were used to analysis of oocysts counts in different groups. Results were expressed as means $\pm$ SE. Probability of values less than 0.05 ($p \leq 0.05$) was considered significant.

**Table 1** The oocysts count per gram of feces (OPG) in experimental infected rabbits from day 4 to day 8 post infection.

| Group | Day 4 PI | Day 5 PI | Day 6 PI | Day 7 PI | Day 8 PI |
|---|---|---|---|---|---|
| Negative Control (NC) | $0.00 \pm 0.00$ | $0.00 \pm 0.00^c$ | $0.00 \pm 0.00^c$ | $0.00 \pm 0.00^c$ | $0.00 \pm 0.00^c$ |
| Positive Control (PC) | $0.00 \pm 0.00$ | $43333.33 \pm 4409.58^a$ | $173666.67 \pm 3282.95^a$ | $380000 \pm 6082.95^a$ | $440666.67 \pm 2962.73^a$ |
| Prebiotic supplemented (PS) | $0.00 \pm 0.00$ | $25166.67 \pm 2743.67^b$ | $107666.67 \pm 4333.33^b$ | $207000 \pm 4618.80^b$ | $262000 \pm 2309.40^b$ |

Notes.
    Data are presented as the means for each group and standard error of the mean (Mean ± SE). a, b,c means within the 4 same column with different superscripts are significantly different at ($P \leq 0.05$).

**Table 2** Prevalence of Enterobactericae infection in examined samples for each group after prebiotic supplementation as a prophylaxis for rabbit intestinal coccidiosis.

| Group | *E. coli* infection | | *Salmonella* infection | | Total prevalence |
|---|---|---|---|---|---|
| | Examined samples | Positive (%) | Examined samples | Positive (%) | |
| Negative Control (NC) | 15 | 3 (20%) | 15 | 1 (6.67%) | 4 (8.89%) |
| Positive Control (PC) | 15 | 6 (40%) | 15 | 3 (20%) | 9 (60%) |
| Prebiotic supplemented (PS) | 15 | 2 (13.33) | 15 | 0 | 2 (13.33%) |
| Total | 45 | 11 (24.44%) | 45 | 4 (8.89%) | 15 |

## RESULTS

**Prebiotic prophylaxis effect on the oocyst count and prevalence of both *E. coli* and *Salmonella* species at day 5 and day 8 post infection**

The oocyst excretion in the feces of infected rabbits began at the day 5$^t$ PI in PS and PC groups. Also, the OPG was significantly reduced in the PS group ($2.51 \times 10^4 \pm 10.36$) when compared with the PC group ($4.33 \times 10^4 \pm 11.52$) (Table 1). This significant ($p \leq 0.05$) decrease in the oocyst count in the PS group continued till the end of the experiment (day 8 PI). Eleven *E. coli* isolates were detected during the present study with overall prevalence of 24.4%. These eleven isolates were three serotypes in NC with prevalence of 20%, six serotypes PC with prevalence of 40% and two serotypes in PS with a prevalence of 13.33% (Table 2). The highest prevalence of *E. coli* was in the PC group while the lowest one was in the PS group. Meanwhile, four *salmonella* serovars were isolated with overall prevalence of 8.89%. The NC group showed only one serovars with a prevalence of 2.22% and the PC group revealed three serovars with a prevalence of 6.67% while prebiotic supplemented group didn't show any *salmonella* isolates (Table 2). The same results of *E. coli* and *Salmonella* prevalence were recorded at day 8 PI.

### The intestinal isolates *E. coli*

Five *E. coli* isolates were recovered from the intestinal tissues and identified as: O78, O125, O152, O115 and O168. The most common isolates were O78 and O152 with prevalence of (27.27%) and they were isolated from jejunum and ileum. While the least common one was O125 (18.18%) which was isolated from ileum (Table 3). The isolates from the NC group were (O152, O152, O168), while the PC group had six serotypes (O168, O152, O125, O125, O78, O78). The PS group showed only two serotypes (O78, O115). The antimicrobial sensitivity tests of *E. coli* serotypes showed high resistance to florfenicol

**Table 3  Grouping of *E. coli* isolates recovered from different organs of prebiotic supplemented rabbits.**

| *E. coli* serogroups | No. of isolates | % | Organs of isolation |
|---|---|---|---|
| O78 | 3 | 27.27 | Jejunum and Ileum |
| O125 | 2 | 18.18 | Ileum |
| O152 | 3 | 27.27 | Jejunum and Ileum |
| O115 | 1 | 9.09 | Caecum. |
| O168 | 2 | 18.18 | Caecum and Ileum |
| Total | 11 | 100 | |

**Table 4  Antimicrobial sensitivity of *E. coli* isolates in examined samples of prebiotic supplemented rabbits.**

| Antimicrobial agents | Resistance | | Intermediate | | Sensitive | |
|---|---|---|---|---|---|---|
| | NO | % | NO | % | NO | % |
| Tetracycline | 9 | 81.81 | 2 | 18.18 | 0 | |
| Sulphamazole | 8 | 72.72 | 3 | 27.27 | 0 | |
| Naldixic acid | 9 | 81.81 | 2 | 18.18 | 0 | |
| Trimethoprim | 9 | 81.81 | 2 | 18.18 | 0 | |
| Gentamycin | 6 | 54.54 | 4 | 36.36 | 1 | |
| Levofloxacin | 8 | 72.72 | 3 | 27.27 | 0 | |
| Florfenicol | 11 | 100 | 0 | 0 | 0 | |
| Amoxicillin | 4 | 36.36 | 6 | 54.54 | 1 | |
| Flumequin | 9 | 81.81 | 1 | 9.09 | 1 | |
| Ciprofloxacine | 4 | 36.36 | 5 | 45.45 | 2 | |
| Amikacin | 1 | 9.09 | 4 | 36.36 | 6 | |
| Neomycin | 11 | 100 | 0 | 0 | 0 | |

and neomycin (100%), tetracycline, naldixic acid, trimethoprim and flumequin (81.81%) (Table 4 & Table S2).

## Prevalence of *Salmonellae* species recovered from intestinal tissue

Three serovars were identified for four isolates; one *Salmonella entrica* subsp. *Enterica* serovar Macclesfield with prevalence of 25%, one *Salmonella entrica* subsp. *Enterica* serovar canada with prevalence of 25% and two *Salmonella entrica* subsp. *enterica* serovar Kisangani with prevalence of 50% (Table 5). *Salmonella* isolates according the organs of isolation showed one *Salmonella entrica* subsp. *enterica* serovar Maccles isolated from caecum, one *Salmonella entrica* subsp. *enterica* serovar Canada isolated from jejunum and two *Salmonella entrica* subsp. *enterica* serovar Kisangani isolated from ileum (Table 5). The antimicrobial sensitivity tests of the isolated *Salmonella* serovars showed higher resistance to sulfamazole, amoxicillin and flumequin (75%) while they were sensitive to levofloxacin and ciprofloxacine (75%) (Table 6 & Table S3).

**Table 5** Prevalence of *Salmonella* serovars isolated from the examined samples of prebiotic supplemented rabbits.

| *Salmonella* serovars | No. of isolates | Group | Organ of isolation | Prevalence % |
|---|---|---|---|---|
| Macclesfield | 1 | PC | Caecum | 25% |
| Canada | 1 | NC | Jujenum | 25% |
| Kisangani | 2 | PC | Ileum | 50% |

**Table 6** Antimicrobial sensitivity of *Salmonella* serovars from the examined samples of prebiotic supplemented rabbits.

| Antimicrobial agents | Resistance | | Intermediate | | Sensitive | |
|---|---|---|---|---|---|---|
| | NO | % | NO | % | NO | % |
| Tetracycline | 2 | 50 | 2 | 50 | 0 | 0 |
| Sulphamazole | 3 | 75 | 1 | 25 | 0 | 0 |
| Naldixic acid | 2 | 50 | 2 | 50 | 0 | 0 |
| Trimethoprim | 0 | 0 | 2 | 50 | 2 | 50 |
| Gentamycin | 1 | 25 | 1 | 25 | 2 | 50 |
| Levofloxacin | 0 | 0 | 1 | 25 | 3 | 75 |
| Florfenicol | 2 | 50 | 2 | 50 | 0 | 0 |
| Amoxicillin | 3 | 75 | 1 | 25 | 0 | 0 |
| Flumequin | 3 | 75 | 1 | 25 | 0 | 0 |
| Ciprofloxacine | 0 | 0 | 1 | 25 | 3 | 75 |
| Amikacin | 1 | 25 | 1 | 25 | 2 | 50 |
| Neomycin | 1 | 25 | 3 | 75 | 0 | 0 |

# DISCUSSION

The use of antibiotics as growth promoters in animals was banned by the European Union Commission (*European Union Commission, 2005*), and since then, prebiotics and probiotics were actively investigated as safe natural alternatives to the antibiotics. Prebiotics were known to have valuable effects on the improvement of the host immune system, productivity and performance in addition to its bactericidal/bacteriostatic activities. The prebiotics were also used as growth promoters in the form of feed additives to increase growth of chickens (*Ashayerizadeh et al., 2009*). It was found that coccidial infection in most cases was associated with secondary bacterial and viral infections that may lead to mortality in infected host (*Taylor et al., 2003*; *Kowalska et al., 2012*).

Therefore, the present study was suggested to investigate the prevalence of bacterial infection in rabbit intestines experimentally infected with coccidiosis and the effect of prebiotic supplementation on it. In the prophylactic trial, the prevalence of *E. coli* was 6.66.11% in the NC group and 13.33% in the PC group while, it was 4.4% in the PS group. These findings were in agreement with those of *Jouany, Gobert & Medina (2008)*, *Kimse (2009)* and *Michelland, Combes & Monteils (2010)*. Prebiotics were proven to have positive effect on certain pathogens where they able to control enteric diseases associated with *E. coli* (*Kritas & Morrison, 2005*; *Timmerman et al., 2005*). In this respect, *Servin & Coconnier (2003)* found that the gram-positive bacterium *Lactobacillus lactis* produced

hydrogen peroxide and reduced of the growth of *Escherichia coli* 0157:H7. In the present study the isolated *E. coli* showed high resistance against neomycin, florfenicol, flumequin, sulphamazole,trimethoprim, nalidixic acid and tetracycline while, they were sensitive to ciprofloxacine and amikacin. These outcomes similar to those of *Flickinger & Fahey (2002)*, *Zhao et al. (2018)* and *Makhol, Habreh & Sakural (2011)*.

On the other hand, the prevalence of *Salmonella* was 2.22% in the NC group and 6.7% in the PC group while it was not detected in the PS group. The presence of *Salmonella* species with coccidiosis was previously reported by *Arakawa et al. (1992)*. They suggested that the infection with *E. tenella* leading to changes in the balance of competitive adherence of bacteria which allowing more colonization of *S. typhimurium* and *Clostridium perfringens*. *Salmonella* isolates reported in the present study showed high resistance to sulphamazole, florfenicol, amoxicillin, flumequin and naldixic acid while it revealed high sensitivity to ciprofloxacine, levofloxacin and gentamycin. Several previous studies have demonstrated antimicrobial sensitivity outcomes similar to those reported in the present study (*Kumar, Sharma & Mani, 2009*; *Camarda et al., 2012*; *Kim et al., 2012*; *Albuquerque et al., 2014*; *Agrawal & Hirpurkar, 2016*; *Lamas et al., 2016*).

The present study showed high prevalence of *E. coli* and *Salmonella* serovars in the PC group that indicates a relation between the infection by the intestinal coccidiosis and the proliferation of enterobacteriacae micro-organisms. This finding was supported by *Szabóová et al. (2012)* as they observed significant decrease in the bacterial prevalence in association with reduction in the *Eimeria* oocysts count in rabbits which were administered a dietary supplementation of natural substances. In addition, *El-Ashram et al. (2020)* found that the coccidiosis infection in post weaning rabbits mostly associated with *E. coli* and *Salmonella* species.

In the present study, the used prebiotic reduced the prevalence of *E. coli* and *Salmonella* species associated with experimentally induced coccidiosis in rabbits with reduction of the adverse effect of coccidiosis. This in agreement with the results of *El-Ashram et al. (2019)* as they found that the prebiotic supplementation reduced the adverse effects of the intestinal coccidiosis in rabbits. Interestingly, *Salmonella* species was absent in the PS group which may reflect the ability of prebiotic to inhibit colonization of *Salmonella*. This finding supported by *Micciche et al. (2018)* as they showed that prebiotic creates of an environment that and inhibit *Salmonella* colonization and growth in chicken intestine. In addition, *Tran et al. (2018)* found that a prebiotic supplementation can cause inhibition to *Salmonella* and *E. coli* infections in pigs. Moreover, *Girgis et al. (2020)* realized that prebiotic (Actigen® a mannan-rich yeast cell wallderived preparation) supplementation diminished the prevalence of *Salmonella* enteritis in cecal contents of layer chickens. Prebiotics are always used as feed additives to improve growth, endorse beneficial gastrointestinal microbiota, and decrease pathogens. The prebiotics increase short chain fatty acid (SCFA) production in the cecum which leading to pathogen reduction (*Micciche et al., 2018*; *Girgis et al., 2020*). The prebiotics promote the overall health and wellbeing of the bird through creation of an intestinal environment unfavorable for *Salmonella* colonization (*Micciche et al., 2018*).

In conclusion, the using of prebiotic as prophylaxis in this study, significantly reduced the prevalence of the *E. coli* and prevented the *salmonella* infection that associated with intestinal coccidiosis in rabbits. In addition, it reduced the intestinal coccidiosis adverse effect as evidenced by the reduction of oocysts shedding.

### Funding

This work was supported by Researcher supporting Project (RSP-2019/3), King Saud University. The funders had no role in study design, data collection and analysis, decision to publish, or preparation of the manuscript.

### Grant Disclosures

The following grant information was disclosed by the authors:
Researcher supporting Project (RSP-2019/3), King Saud University.

### Competing Interests

The authors declare that there are no competing interests.

### Author Contributions

- Shawky M. Aboelhadid and El-Sayed Abdel-Kafy conceived and designed the experiments, analyzed the data, authored or reviewed drafts of the paper, and approved the final draft.
- Shaymaa Hashem and Lilian N. Mahrous performed the experiments, analyzed the data, prepared figures and/or tables, and approved the final draft.
- Eman M. Farghly conceived and designed the experiments, authored or reviewed drafts of the paper, microbiological work, and approved the final draft.
- Abdel-Azeem S. Abdel-Baki and Saleh Al-Quraishy analyzed the data, authored or reviewed drafts of the paper, and approved the final draft.
- Asmaa A. Kamel performed the experiments, prepared figures and/or tables, and approved the final draft.

### Animal Ethics

The following information was supplied relating to ethical approvals (i.e., approving body and any reference numbers):

This experiment was conducted under the roles of the ethical standards approved by Faculty of Veterinary Medicine, Beni-Suef University, Egypt and its specific approval number was (BSUV-39/2019).

### Data Availability

Raw data is available in the Supplemental Files.

## Supplemental Information

Supplemental information for this article can be found online at http://dx.doi.org/10.7717/peerj.10714#supplemental-information.

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
