# Peer review of "Prebiotic supplementation effect on *Escherichia coli* and *Salmonella* species associated with experimentally induced intestinal coccidiosis in rabbits"

_PeerJ, doi:10.7717/peerj.10714_

## Round 0.1 · original submission · Major Revisions

Thanks for considering PeerJ for your manuscript, Please address all the comments by the reviewers, and discuss why the sampling time was halted at 8 days post-infection as alluded by the reviewers. And how does this impacts the findings? In addition, several details about the diet and other the composition of the probiotic used must be included.

Also there multiple spelling and grammatical mistakes that need to be polished before the manuscript is ready for publication. Please proofread the manuscript to address these issues.

·

Basic reporting

Add some detail about coccidiosis their prevalence and than link with bacterias and antibiotic resistant.

Please add uptodate reference (is the major cause of antibiotic resistance (WHO 2000)

can you add composition of prebiotic supplements available in field and their effects and possible role in antibiotic resistant

Experimental design

Why you only choosed male rabbit?

Please add composition of diet it is must to know because diet is potential risk of coccidiosis

Please add name of antibiotics which was used in 12 antibiotic discs and give reference

Validity of the findings

Study is some how interesting and results are looking good, but it need some details before making final decision

Additional comments

Please response above mention question and try to add latest reference

Reviewer 2 ·

Basic reporting

The manuscript entitled “Prebiotic supplementation effect on Escherichia coli and Salmonella species associated with experimentally induced intestinal coccidiosis in rabbits” describes the effect of prebiotic supplementation on the presence of Escherichia coli and Salmonella species in rabbits experimentally infected with intestinal coccidiosis was investigated. I found the topic of this study interesting in the Egyptian context and fitting in the scope of the journal. The study is well executed and conducted and the manuscript is well written.

Experimental design

The study is well executed and conducted.

Validity of the findings

The findings are an added value to the scientific community.

Additional comments

Well-done!

Reviewer 3 ·

Basic reporting

no comment

Experimental design

The prepatent period for E. magna, E. flavescens and E. intestinalis is much longer than 4 days, and in turn the oocyst shedding peaks are around 8-10 days PI. The most severe period for the coccidiosis, mainly with secondary bacterial infection, usually appears since the shedding of oocysts. For such situation, Trials with more animals and sampling at more time points after 8 day PI would be valuable for the evaluation.

Validity of the findings

no comment

Additional comments

the oocyst output in Table 1 could not fully demonstrate the prophylactic effect of the product, as the oocyst shedding peaks are around 8-10 days PI.
the number of E.coli and Salmonella subtypes detected in this study reveals the severity of the bacterial infection, but not all of it, the infection intensity would be in consideration.

---

## Round 0.2 · Minor Revisions

Thanks for making all the recommended changes and answering the reviewer's questions. I think the manuscript is almost ready for publication, the reason I am sending it back for minor revisions is there are still multiple spelling and grammatical mistakes spread in the article making it hard to interpret at times, below are few I came across. Please take this opportunity to re-read through the manuscript make any changes and proof read it multiple times and improve the overall quality of the text.

Line 72 : rabbitaries -> rabbitries
Line 80 : non enterotoxigenic -> non-enterotoxigenic
91: inLactobacillus -> in Lactobacillus
136: Cervical dislocation was done due to they were not of heavy weight -> Cervical dislocation was done because they were not of heavy weight
162: ocysts or oocysts?
185: Remove comma after while
226: space needed after comma

---

## Round 0.3 · accepted · Accept

Thanks for making those last edits to improve the manuscript text, and for considering PeerJ to publishing this important work. Congratulations !